# Combination of Biochar with N–Fertilizer Affects Properties of Soil and N₂O emissions in Maize Crop

Tatijana Kotuš [1], Vladimír Šimanský [2], Katarína Drgoňová [1], Marek Illéš [1], Elżbieta Wójcik-Gront [3], Eugene Balashov [4], Natalya Buchkina [4], Elena Aydın [1] and Ján Horák [1,*]

[1] Institute of Landscape Engineering, Faculty of Horticulture and Landscape Engineering, Slovak University of Agriculture, 949 76 Nitra, Slovakia; xkotus@uniag.sk (T.K.); katarina.drgonova@uniag.sk (K.D.); marek.illes@zsvs.sk (M.I.); elena.aydin@uniag.sk (E.A.)

[2] Department of Soil Science, Institute of Agronomic Sciences, Faculty of Agrobiology and Food Resources, Slovak University of Agriculture, 949 76 Nitra, Slovakia; vladimir.simansky@uniag.sk

[3] Department of Biometry, Institute of Agriculture, Warsaw University of Life Sciences—SGGW, Nowoursynowska 159, 02-776 Warsaw, Poland; elzbieta_wojcik_gront@sggw.edu.pl

[4] Department of Soil Physics, Physical Chemistry and Biophysics, Agrophysical Research Institute, Grazhdansky pr. 14, 195220 St. Petersburg, Russia; eugene_balashov@mail.ru (E.B.); buchkina_natalya@mail.ru (N.B.)

* Correspondence: jan.horak@uniag.sk

**Abstract:** One of the challenges of harnessing higher productivity levels and sustainability of agriculture related to N fertilization is in expanding soil N₂O emissions, which has become a serious issue in recent years. Recent studies suggest that biochar may be the solution to this problem, but there is still a knowledge gap related to biochar application rates and its reapplication in Central Europe; therefore, in this study, we investigated the effect of biochar (initial application and reapplication in 2014 and 2018, respectively, at rates of 0, 10 and 20 t ha$^{-1}$) combined with N-fertilizer (N0—0 kg N ha$^{-1}$; N1—108 kg N ha$^{-1}$ and N2—162 kg N ha$^{-1}$) during the growing season of maize in 2019 (warm temperature with normal precipitation) on the changes of soil properties and N₂O emissions in the silty loam, Haplic Luvisol, in the temperate climate of Slovakia. The results showed that the application and reapplication of biochar proved to be an excellent tool for increasing soil pH (in the range 7–13%), soil organic carbon—C$_{org}$ (2–212%), and reducing the soil's NH₄⁺ (41–69%); however, there were more pronounced positive effects when biochar was combined with N-fertilizer at the higher level (N2). The same effects were found in the case of N₂O emissions (reduction in the range 33–83%). Biochar applied without N-fertilizer and combined with the higher fertilizer level had a suppressive effect on N₂O emissions. Biochar did not have any effect on maize yield in 2019.

**Keywords:** chemical properties of soil; greenhouse gas emissions; Luvisol; soil organic carbon

## 1. Introduction

Anthropogenic greenhouse gas (GHG) emissions have increased since the pre-industrial era, driven by economic and population growth; nowadays, they are higher than ever [1]. Sectors such as transport, energy production, industry, agriculture, and land–use change are significant sources of GHG emissions, and they contribute 24% of global anthropogenic GHG emissions [2]. These gases are released as a result of different agricultural management practices, which impact soil respiration, nitrification, and denitrification processes [3–5]. For example, the concentration of N₂O in the atmosphere has increased by more than 20% since the industrial revolution due to anthropogenic activities [1]. N₂O is long-lived greenhouse gas with global warming potential that is 298 times higher than CO₂ for a time horizon of 100 years [6]. Most N₂O that is emitted from the soil is produced by three biotic processes: denitrification [7], nitrification [8], and nitrifier denitrification [9]. In recent decades, various strategies have been sought to eliminate the negative effects of

agriculture. Several studies suggest that biochar may be the solution to the issue mentioned above. Biochar is an aromatic substance that is produced at high temperatures (350–600 °C) by pyrolysis of different feedstocks [10]. The physical and chemical properties of biochar depend on pyrolysis temperature and feedstock type [11,12]. Biochar has a large surface area and porous carbonaceous structure [13]. Due to the physical and chemical properties of biochar, biochar application in soils has been advocated as a potential management strategy to improve physical [14–18] and chemical properties of soil [19–23], increase crop yields [24–26], and reduce GHG emissions from soils through various mechanisms [27–29].

Biochar suppressive effects on soil $N_2O$ emission has been observed not only in short-term laboratory studies [30,31], but also in long-term field experiments [32,33]. The major factors controlling $N_2O$ emissions from agricultural soils include being dissolved in organic carbon content, soil N availability, temperature, soil moisture, and soil pH [34]. Biochar has an impact on soil $N_2O$ emissions, and this impact is dependent on the biochar rate. For example, Castaldi et al. [35] found that adding 3 t ha$^{-1}$ and 6 t ha$^{-1}$ of biochar to an agricultural soil could effectively reduce soil $N_2O$ emissions by 26% and 79%, respectively. The reduction effect of high biochar rates on soil $N_2O$ emissions is usually more pronounced. In addition, high biochar rates can adsorb more $NO_3^-$ in the soil, thus reducing the substrate for denitrification [36]. High biochar rates can increase the number of *Nitrospira* bacteria in soils, thus inhibiting the generation of $N_2O$ in the soil [37]; however, the effects of high biochar rates on soil $N_2O$ emissions can decrease with time, and sometimes biochar application can even support soil $N_2O$ emissions. Therefore, the effect of biochar on the reduction of $N_2O$ emissions from soils is closely related to the applied rate of biochar [38]. The effect of biochar decreases with time, most likely due to decomposition, utilization, and consumption of biochar nutrients [39]. Most studies report reductions in $N_2O$ emissions after biochar amendments, but the results sometimes differ depending on the type of experiment (laboratory soil incubations, pot experiments, or field experiments), the duration (short, mid, or long–term experiments), the type of soil, and biochar, or the driving soil mechanism leading to $N_2O$ formation and release. Several metanalyses [40–42] deal and sum up previous evidence of these diverse observed effects.

As mentioned above, the potential of biochar applications for mitigating GHG emissions from soils across the world is evident; however, the influence of different nitrogen fertilizer levels on the $N_2O$ emissions of biochar-amended soils requires investigation. Particularly little information is available on the response of soil $N_2O$ emissions and soil properties to the changes in initial and reapplied rates of biochar with or without N fertilizer; therefore, we investigated the changes in the soil properties and $N_2O$ emissions from biochar amended soils (after initial application and reapplication) as affected by different levels of N-fertilization. We hypothesized that nitrogen emissions would increase due to N fertilization and decrease as a result of the biochar application, with temperature and water content in the soil being the most significant variables affecting $N_2O$ emissions. We also assumed that the effects would be more pronounced at higher doses of both N fertilizer and biochar, and that biochar reapplication would be more effective than the initial biochar application.

## 2. Materials and Methods

### 2.1. Description of the Experimental Site

The experimental field was established in 2014, in Malanta, located in the Nitra region, Slovakia, with a latitude of 48°19′04′′ N and longitude of 18°08′45′′ E. The study area has a warm temperate climate, fully humid with warm summers (Cfb), according to the classification by Köppen-Geiger et al. [43]. The mean annual air temperature was 10.8 °C, whereas the mean annual sum of precipitation was 559 mm (based on 30-year climatic normal from 1991 to 2020). In 2019, the annual sum of precipitation was 625.4 mm, and the mean air temperature was 10.9 °C (Table 1). The soil at the experimental site was classified as Haplic Luvisol, with a silty loam texture, containing an average of 15.2% of sand, 59.9%

of silt, and 24.9% of clay. The average soil organic carbon content before the commencement of the field experiment was 9.13 g kg$^{-1}$, and the soil pH in KCl was 5.71.

**Table 1.** Evaluation of monthly precipitation and mean air temperature in 2019 compared with the 30-year climatic normal (1991–2020).

| | Air Temperature | | | Precipitation | | |
|---|---|---|---|---|---|---|
| **Month** | **Mean (°C)** | **Deviation from the Normal (°C)** | **Description** | **Total (mm)** | **% of Normal** | **Description** |
| January | −3.5 | −3.0 | cold | 54.8 | 166.1 | very wet |
| February | +0.9 | −0.4 | normal | 27.4 | 94.5 | normal |
| March | +5.0 | −0.5 | normal | 22.4 | 67.9 | dry |
| April | +9.4 | −2.0 | cold | 21.4 | 59.4 | dry |
| May | +9.3 | −6.7 | extremely cold | 134.8 | 228.5 | extremely wet |
| June | +18.7 | −0.9 | normal | 29.0 | 49.2 | very dry |
| July | +18.0 | −3.7 | extremely cold | 52.2 | 80.3 | normal |
| August | +18.4 | −2.7 | very cold | 64.0 | 116.4 | normal |
| September | +12.6 | −3.3 | normal | 52.8 | 91.0 | normal |
| October | +8.7 | −1.7 | cold | 17.8 | 38.7 | very dry |
| November | +5.0 | −0.6 | normal | 95.4 | 212.0 | extremely wet |
| December | −0.1 | −0.8 | normal | 53.4 | 127.1 | wet |

### 2.2. Characterization of the Treating Materials: Biochar and Nitrogen Fertilizer

The biochar used in this study was produced by the pyrolysis of paper fiber sludge and a grain husks mixture (in a ratio of 1:1) at 550 °C for 30 min in a Pyreg reactor (Pyreg GmbH, Dörth, Germany). The physical and chemical properties of the biochar are shown in Table 2. Commercially available calcium–ammonium nitrate was used as N-fertilizer for the current study.

**Table 2.** Basic physical and chemical properties of the biochar used in the study (Certificate Nr. 1013069).

| Parameter | Average Value | Methods of Determination | Unit |
|---|---|---|---|
| Range of particle size | 1–5 | Laser diffraction | mm |
| Bulk density | 0.21 | calculated | g cm$^{-3}$ |
| Specific surface area | 21.7 | DIN 66132/ISO 9277 | m$^2$ g$^{-1}$ |
| pH value (KCl) | 8.8 | DIN ISO 10390 | – |
| Ash content | 38.3 | Analog DIN 51719 | % |
| Carbon (C) content | 53.1 | DIN 51732 | % |
| Nitrogen (N) content | 1.4 | DIN 51732 | % |
| C:N ratio | 37.9 | calculated | – |
| H/C | <0.6 | calculated | – |
| H/C$_{org}$ | <0.7 | calculated | – |
| O/C | <0.4 | calculated | – |

### 2.3. Design of the Experimental Plots and Treatments

The study was carried out in the field from April to October of 2019. The biochar was first applied in this experimental field in 2014 and reapplied again in 2018. The experimental setup included treatments with or without biochar and with a low or high rate of mineral N-fertilizer. The initial biochar treatments (in three replicates) included three biochar rates—0 (B0), 10 (B10) and 20 (B20) t ha$^{-1}$—in 2014 which were applied to the plots of 4 × 6 m. All the experimental plots were separated by protection rows that were 0.5 m wide. In 2018, the initial plots were divided into two subplots (4 × 3 m) which were labeled as A and B. All the B subplots were amended with the biochar for the second time (biochar reapplication) at the same rates as in 2014. Three different rates of the mineral N-fertilizer (N0, N1, and N2) were applied to the soil of the experiment every year, depending on the planted crop. In 2019, N-fertilizer was applied at the rates of 0, 108,

and 162 kg N ha$^{-1}$. The doses from 2014 up to 2018 were as follows, in order of the N0, N1, and N2 levels: 2014—0, 40, 80 kg N ha$^{-1}$; 2015—0, 160, 240 kg N ha$^{-1}$; 2016—0, 100, 150 kg N ha$^{-1}$; 2017—0, 160, 240 kg N ha$^{-1}$; 2018—0, 40, 80 kg N ha$^{-1}$. Figure 1 shows 27 plots of the initial field experiment, which were used for the study in 2019: A—subplots with the biochar applied in 2014 only; B—subplots with the biochar reapplied in 2018; and plots with control treatments without the biochar but with different rates of the mineral fertilizer. Crop rotation from the very first experiment set up with biochar in 2014 was as follows: spring barley (*Hordeum vulgare* L.) (2014), maize (*Zea mays* L.) (2015), spring wheat (*Triticum aestivum* L.) (2016), maize (2017), and once again, spring barley in 2018. In 2019, maize (variety LG 3.315) was planted at the beginning of April and harvested at the beginning of October.

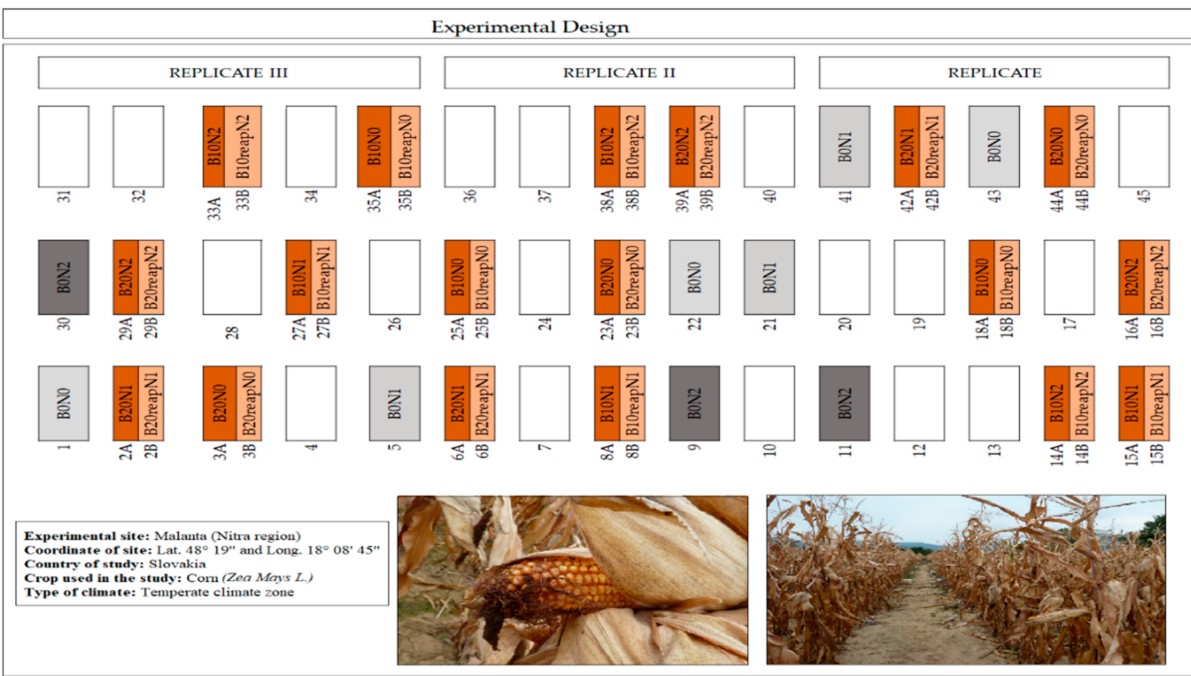

**Figure 1.** Schematic arrangement of the experimental site with 27 highlighted plots which were used in this study: A—subplots with the biochar applied in 2014 only; B—subplots with the biochar reapplied in 2018; and plots with control treatments without the biochar but with different rates of the mineral fertilizer.

*2.4. Soil Sampling and Analyses*

All soil sample analysis methods used in frame of this study are listed in Hrivňáková et al. [44]. Soil water content (SWC) in the 0–0.1m soil layer was determined twice a month (from April to October in 2019) gravimetrically by drying the soil at 105 °C. The soil temperature was measured at a depth of 0.05 m by an electronic thermometer (Volcraft DET3R) twice a month. Disturbed soil samples, collected once a month from April to October 2019, were used to study particular physicochemical and agrochemical parameters of the soil with different treatments. The pH values of the soil were measured using the potentiometric method in 1 mol L$^{-1}$ KCl (ratio of soil to KCl, 1:2.5) using a pH meter (HANNA instruments HI 2211). The concentrations of nitrate (NO$_3^-$) and ammonium (NH$_4^+$) in the soil were measured by a spectrophotometer (Spectroflex 6100). Soil reaction (pH$_{(KCl)}$) as well as mineral nitrogen were measured once a month from April to October. C$_{org}$ content in the samples collected in April and September was measured with the Tyurin wet oxidation method [45]. Bulk density was determined in undisturbed soil samples (100 cm$^3$) gravimetrically in the spring and autumn of 2019; both times, 3 undisturbed soil samples were taken from each individual plot, which makes 9 representative soil samples per treatment (a grand total of 135 soil samples per sampling event).

### 2.5. Gas Sampling and Analysis

Gas samples from the studied agricultural soil for $N_2O$ emission measurements were collected every two weeks. The closed chamber technique was used for gas sample collection [46]. The sampling chamber consisted of two parts: a metal collar frame (galvanized sheet) at the bottom (depth 0.10 m) and removable cover PVC chamber (diameter 0.3 m, height 0.25 m). The metal collar frame was inserted directly into the soil. The removable chamber was placed on the top of the collar during the gas sampling occasions and removed afterwards. An air-tight 60 mL plastic syringe was used to collect gas samples through chamber tube fittings (20 mL, sealed with a septum) at 0, 30, and 60 min after its placing on collar frames followed by water-sealing. Then, gas samples were transferred to 12 mL pre-evacuated glass vials (Labco Exetainer). A gas chromatograph (GC 2010 Plus, Shimadzu Corporation, Kyoto, Japan) equipped with an electron capture detector (ECD) was employed for measuring $N_2O$ concentration in the gas samples in ppb. Cumulative $N_2O$ emissions were calculated by plotting daily fluxes through time, interpolating linearly between them, and integrating the area under the curve [47].

### 2.6. Maize Yield Determination

Whole plants of maize were manually sampled at the end of the vegetation season on 9 October 2019 from randomly selected 1 m long continuous rows on each plot. In the laboratory, the cobs were separated from the rest of aboveground biomass and counted. After that, kernels were manually separated from cobs, counted, oven-dried, and weighed. Grain yield was calculated as the multiplication of total number of grains per ear, average grain weight, and the total number of ears per area unit ($m^2$) [48].

### 2.7. Statistical Analyses

Statistical analyses were performed using STATISTICA software ver. 13.1 [49]. A one–way analysis of variance (ANOVA) was used to evaluate the effects of different biochar rates on the measured parameters. The Tukey procedure was used for daily mean $N_2O$ emissions as well as soil parameters and crop yield differences between the treatment means. Multiple linear regression analysis was performed for the variables that influence the $N_2O$ emission, and Principal Component Analysis (PCA) was carried out for the treatments, and all the measured variables separately, for the two soil sampling/measurement occasions (spring and autumn) in 2019.

## 3. Results and Discussion

### 3.1. Soil Chemical Properties

The initial soil pH (before the experiment was established) was slightly acidic (pH 5.6). Biochar applied to the soil with or without N-fertilizer significantly increased ($p < 0.05$) the soil pH in all the treatments (except for B10N0 and B20N1 treatments), compared with the control treatments without biochar. The soil pH rise was in the range of 0.23–0.38, 0.32–0.63, and 0.50–0.72 pH units for N0, N1, and N2 fertilization levels, respectively. Similar results, with the application of 10, 20, and 30 t ha$^{-1}$ biochar to Haplic Luvisol, were published by another study [50]. These authors stated that in their short-term as well as long-term experiments, the initial application of biochar significantly increased the soil pH, whereas a more significant increase was linked with the application of a higher rate. Since the biochar is alkaline (our case biochar pH 8.8—Table 2), it might increase the soil pH, but this effect could be softened by adding nitrogen fertilization. For example, Yu et al. [51] compared the effects of N ammonium fertilizer alone, at a rate of 30 kg N$^{-1}$ ha$^{-1}$, but they also compared the effect of its combination with 19.5 t ha$^{-1}$ of biochar. After the addition of N alone, the soil pH value was 5.3, and the pH increased from 5.3 to 6.0 after the application of biochar with N. Long-term use of nitrogen fertilizers in the ammonium form could be a reason for soil acidification. In this study, the effect of acidification with N fertilization was eliminated by using calcium–ammonium nitrate, and the maximum rise in soil pH (Table 3) was noted in all the treatments, with reapplied biochar at both rates (B10reap and

B20reap), combined with all N-fertilizer levels. One of the reasons for the observed results might be the oxidation of the biochar surface over time, which creates more carboxylic groups [52], but the main reason why pH in the soil was higher after biochar application is the liming effect of biochar itself [53]. Biochars generally have a higher pH (same as in our case) as a result of the production process. During biochar production, acid functional groups of organic feedstocks are removed and the ash content increases, causing biochar to be more basic than the material it is being produced from [54]. A soluble form of ash alkalinity in biochar can be rapidly released into the soil and leached down the soil profile to alter soil pH. Thus, biochars can be used as an alternative ameliorant to neutralize soil acidity [21,55].

**Table 3.** Selected chemical properties (soil pH, $NH_4^+$, $NO_3^-$, SOC) of the soil after different treatments in 2019 (means $\pm$ standard errors). Different letters ([a], [b]) within columns for each fertilized group indicate that the means are significantly different at $p < 0.05$ according to the Tukey procedure.

| Treatments | pH $_{(KCl)}$ | $NH_4^+$ (mg kg$^{-1}$) | $NO_3^-$ (mg kg$^{-1}$) | $C_{org}$ (Spring) (g kg$^{-1}$) | $C_{org}$ (Autumn) (g kg$^{-1}$) |
|---|---|---|---|---|---|
| | $n = 3$ | $n = 3$ | $n = 3$ | $n = 3$ | $n = 3$ |
| Not fertilized group—N0 level (0 kg N ha$^{-1}$) | | | | | |
| B0N0 | 5.21 $\pm$ 0.2 [a] | 15.39 $\pm$ 10.1 [a] | 18.78 $\pm$ 7.5 [a] | 12.00 $\pm$ 0.3 [a] | 14.29 $\pm$ 2.5 [ab] |
| B10N0 | 5.21 $\pm$ 0.2 [a] | 5.55 $\pm$ 0.5 [a] | 14.49 $\pm$ 1.5 [a] | 12.26 $\pm$ 2.1 [a] | 10.67 $\pm$ 2.4 [a] |
| B20N0 | 5.51 $\pm$ 0.2 [b] | 6.05 $\pm$ 1.0 [a] | 12.09 $\pm$ 1.9 [a] | 14.22 $\pm$ 1.5 [a] | 11.63 $\pm$ 1.6 [ab] |
| B10reapN0 | 5.44 $\pm$ 0.1 [ab] | 5.43 $\pm$ 0.8 [a] | 12.25 $\pm$ 2.0 [a] | 16.57 $\pm$ 2.6 [a] | 12.43 $\pm$ 2.2 [ab] |
| B20reapN0 | 5.59 $\pm$ 0.1 [b] | 5.71 $\pm$ 1.2 [a] | 11.46 $\pm$ 2.8 [a] | 17.02 $\pm$ 2.9 [a] | 20.83 $\pm$ 1.2 [b] |
| Fertilized group—N1 level (108 kg N ha$^{-1}$) | | | | | |
| B0N1 | 4.81 $\pm$ 0.2 [a] | 42.05 $\pm$ 18.2 [a] | 28.20 $\pm$ 7.1 [a] | 9.51 $\pm$ 0.8 [a] | 10.10 $\pm$ 0.7 [a] |
| B10N1 | 5.35 $\pm$ 0.5 [ab] | 24.94 $\pm$ 8.2 [a] | 25.85 $\pm$ 6.6 [a] | 13.98 $\pm$ 2.2 [a] | 14.90 $\pm$ 2.7 [ab] |
| B20N1 | 5.13 $\pm$ 0.2 [ab] | 28.87 $\pm$ 9.1 [a] | 26.91 $\pm$ 6.7 [a] | 15.68 $\pm$ 1.4 [a] | 14.93 $\pm$ 1.5 [ab] |
| B10reapN1 | 5.44 $\pm$ 0.6 [ab] | 41.87 $\pm$ 11.2 [a] | 27.41 $\pm$ 7.5 [a] | 13.39 $\pm$ 1.8 [a] | 13.23 $\pm$ 0.7 [ab] |
| B20reapN1 | 5.25 $\pm$ 0.1 [b] | 35.81 $\pm$ 7.3 [a] | 32.15 $\pm$ 5.1 [a] | 17.22 $\pm$ 2.8 [a] | 18.80 $\pm$ 2.2 [b] |
| Fertilized group—N2 level (162 kg N ha$^{-1}$) | | | | | |
| B0N2 | 4.40 $\pm$ 0.1 [a] | 80.19 $\pm$ 14.6 [b] | 28.53 $\pm$ 8.1 [a] | 9.94 $\pm$ 0.8 [a] | 10.31 $\pm$ 2.1 [a] |
| B10N2 | 4.90 $\pm$ 0.3 [b] | 25.11 $\pm$ 11.0 [a] | 30.20 $\pm$ 7.5 [a] | 14.98 $\pm$ 1.6 [ab] | 9.43 $\pm$ 2.6 [a] |
| B20N2 | 4.97 $\pm$ 0.3 [b] | 41.68 $\pm$ 14.4 [a] | 33.97 $\pm$ 8.1 [a] | 15.96 $\pm$ 0.8 [ab] | 13.80 $\pm$ 2.7 [a] |
| B10reapN2 | 5.09 $\pm$ 0.4 [b] | 34.49 $\pm$ 20.7 [a] | 34.82 $\pm$ 9.0 [a] | 17.07 $\pm$ 1.3 [ab] | 13.77 $\pm$ 1.2 [a] |
| B20reapN2 | 5.12 $\pm$ 0.2 [b] | 30.05 $\pm$ 8.2 [a] | 35.62 $\pm$ 3.6 [a] | 21.11 $\pm$ 2.7 [b] | 17.30 $\pm$ 3.8 [a] |

Note: $C_{org}$–soil organic carbon.

During the growing season of 2019, N-fertilization resulted in a significant increase of the average $NH_4^+$ content in the studied soil from 15.39 $\pm$ 10.1 mg kg$^{-1}$ in B0N0 treatment, to 42.05 $\pm$ 18.2 mg kg$^{-1}$ and 80.19 $\pm$ 14.6 mg kg$^{-1}$ in B0N1 and B0N2 treatments, respectively (Table 3). The average content of $NH_4^+$ in the soil within the fertilization groups significantly ($p < 0.05$) decreased after biochar application and replication for N2 treatments compared with the group controls (B0N2). The most pronounced reduction in the average soil $NH_4^+$ content was found at the N2 fertilization level where the average soil $NH_4^+$ content was the highest and was reduced by 41–69% as a result of biochar application or reapplication. Zhang et al. [56] also stated that the $NH_4^+$ content after biochar addition decreased by 10.3–18.5% when compared with control treatments. These results are also in line with the results of Liu et al. [38], who found that after biochar application, the content of $NH_4^+$ in the soil was significantly reduced. For the N0 fertilization level, the reduction was not significant and very low, similarly to the average soil $NH_4^+$ content. For the N1 fertilization level, the reduction in the average soil $NH_4^+$ content was not significant. According to the scientific literature, biochar can improve soil cation exchange capacity due to the acid functional groups present on its surfaces, which can result in extra $NH_4^+$

absorption [57]. In soils, ammonia can be very rapidly oxidized to $NO_3^-$ produced by microorganisms in the nitrification process, as well as lost through the volatilization process or leaching.

During the growing season of 2019, the average soil $NO_3^-$ content was slightly, but insignificantly, higher in N1 and N2 fertilization groups compared with the N0 treatment. In the N0, N1, and N2 fertilization groups, the average values of the soil $NO_3^-$ content did not change significantly after biochar application or reapplication in either rate, and ranged from 25.85 to 32.15 mg $kg^{-1}$ for N1, and from 28.53 to 35.62 mg $kg^{-1}$ for N2. Zhang et al. [56] also noted a gradual comparison with our data, that showed a statistically significant reduction in $NO_3^-$ due to increasing rates (10, 20 and 40 t $ha^{-1}$) of biochar, with a decrease in the range of 12.6–44.2% during the study period of 2017–2020. Biochar, being negatively charged, has very low anion exchange capacity, the mechanism of $NO_3^-$ adsorption by biochar in soils is more likely to be based on physical principles such as a high specific surface area [58] and increased micropore volume [59], which was probably the case in the N0 fertilization group in our experiment. Soil $NO_3^-$ is more easily lost through leaching and runoff to groundwater, or it acts as a precursor to $N_2O$ emission in the atmosphere [60]. Sial et al. [61] reported that biochar, in their experiment, was raising the soil adsorption of $NO_3^-$, which was reducing the N content available for microorganisms, and thus, was reducing $N_2O$ emissions; however, this was not the case for the fertilized treatments in our experiment.

In both sampling times (spring, autumn), the $C_{org}$ content was higher after the application of biochar in all fertilization treatments (N0, N1 and N2); however, the significant changes in the $C_{org}$ content were observed only after the higher rate of biochar reapplication in the N0 and N1 fertilized treatments in autumn, and due to the reapplication of higher rates of biochar in spring, but only after the fertilized treatments took place (N2). Such trends were observed in this experiment from the beginning [62], but also throughout its duration [63]. The same results mean that the increase of $C_{org}$ and C sequestration, as an effect of a higher biochar rate (20 t $ha^{-1}$); these results were published in a study by Zhang et al. [56]. Moreover, Cross et al. [64] reported that the application of 20 t of biochar $ha^{-1}$ increase $C_{org}$ by approximately 0.5%, and this effect remains for much longer in soils, than, for instance, compost carbon. Biochar is a significant source of stable C [17,18,22], and its application into the soil can be linked with decreasing microbial activity, lower $CO_2$ production, and a fall in mineralization. Biochars, in relation to their properties, can provoke positive or negative priming effects, which can be affected by added N fertilization [18], as was the case in this study. In addition to the supplied C through biochar and N fertilization, microbial biomass, root exudate production, and overall underground biomass increased, which was reflected in the higher $C_{org}$ content in the soil in these treatments. A higher content of $C_{org}$ indicates carbon sequestration in the soil. This observation is in line with many other scientific studies [65–67].

### 3.2. Soil Physical Properties

During the growing season of 2019, the soil temperature ranged between 9.3 and 18.7 °C, with precipitation ranging between 17.8 and 134.8 mm (Table 1). Moreover, in terms of air temperature and precipitation, the growing season was cold and normal, respectively. As shown in Table 4, biochar application to the soil had no effects on the average values of soil temperature and SWC, despite some opposing information being published in the scientific literature. In the treatments with biochar application and reapplication at the rates of 10 and 20 t $ha^{-1}$, and at different levels of fertilization N0, N1, and N2, soil temperature ranged from +0.11 to +1.42, from −0.32 to −0.14, and from +0.28 to +0.61 °C, compared with the corresponding controls (B0N0, B0N1, and B0N2). Liu et al. [38] found that biochar treatments increased soil temperature on average, from 4.4% to 14.4%. According to Ding et al. [68], biochar may help stabilize the fluctuation of the soil temperature and increase soil temperature accumulation. Biochar application changes the soil color, and thus, can have an influence on soil temperature. After biochar application to the soil, at a

rate of 30–60 t ha$^{-1}$, biochar can reduce the soil albedo by 80% [69]. Maize is a crop that grows very rapidly, and the soil surface is very rapidly covered and protected from direct sunlight by the biomass. This must have been the main reason why there was no effect of the biochar on the average soil temperature in our experiment.

**Table 4.** Effect of biochar application on selected soil physical properties (soil temperature, bulk density, soil water content) measured during the growing season of 2019 (means ± standard errors). Different letters ($^a$), within columns for each fertilized group, indicate that treatment means are significantly different at $p < 0.05$, according to the Tukey procedure.

| Treatments | Soil T (°C) | BD (Spring) (g cm$^{-3}$) | BD (Autumn) (g cm$^{-3}$) | SWC (% Mass) |
|---|---|---|---|---|
| | $n = 3$ | $n = 3$ | $n = 3$ | $n = 3$ |
| Not fertilized group—N0 level (0 kg N ha$^{-1}$) | | | | |
| B0N0 | 18.60 ± 0.4 $^a$ | 1.38 ± 0.01 $^a$ | 1.48 ± 0.02 $^a$ | 14.63 ± 0.8 $^a$ |
| B10N0 | 18.87 ± 0.3 $^a$ | 1.37 ± 0.06 $^a$ | 1.50 ± 0.04 $^a$ | 14.04 ± 0.7 $^a$ |
| B20N0 | 18.71 ± 0.3 $^a$ | 1.27 ± 0.01 $^a$ | 1.49 ± 0.03 $^a$ | 15.23 ± 0.9 $^a$ |
| B10reapN0 | 19.02 ± 0.3 $^a$ | 1.38 ± 0.04 $^a$ | 1.52 ± 0.03 $^a$ | 14.23 ± 0.6 $^a$ |
| B20reapN0 | 18.97 ± 0.3 $^a$ | 1.28 ± 0.04 $^a$ | 1.48 ± 0.04 $^a$ | 15.20 ± 1.0 $^a$ |
| Fertilized group—N1 level (108 kg N ha$^{-1}$) | | | | |
| B0N1 | 18.74 ± 0.3 $^a$ | 1.40 ± 0.03 $^a$ | 1.49 ± 0.03 $^a$ | 12.85 ± 0.7 $^a$ |
| B10N1 | 18.60 ± 0.5 $^a$ | 1.32 ± 0.03 $^a$ | 1.53 ± 0.04 $^a$ | 14.04 ± 1.0 $^a$ |
| B20N1 | 18.42 ± 0.3 $^a$ | 1.35 ± 0.05 $^a$ | 1.48 ± 0.07 $^a$ | 14.52 ± 0.9 $^a$ |
| B10reapN1 | 18.57 ± 0.4 $^a$ | 1.30 ± 0.02 $^a$ | 1.51 ± 0.04 $^a$ | 14.09 ± 0.7 $^a$ |
| B20reapN1 | 18.51 ± 0.3 $^a$ | 1.32 ± 0.03 $^a$ | 1.48 ± 0.03 $^a$ | 15.07 ± 1.1 $^a$ |
| Fertilized group—N2 level (162 kg N ha$^{-1}$) | | | | |
| B0N2 | 18.38 ± 0.3 $^a$ | 1.40 ± 0.03 $^a$ | 1.52 ± 0.04 $^a$ | 14.15 ± 1.3 $^a$ |
| B10N2 | 18.84 ± 0.4 $^a$ | 1.35 ± 0.03 $^a$ | 1.49 ± 0.04 $^a$ | 13.99 ± 1.3 $^a$ |
| B20N2 | 18.66 ± 0.2 $^a$ | 1.33 ± 0.03 $^a$ | 1.38 ± 0.07 $^a$ | 14.77 ± 0.8 $^a$ |
| B10reapN2 | 18.81 ± 0.4 $^a$ | 1.30 ± 0.04 $^a$ | 1.44 ± 0.01 $^a$ | 14.37 ± 1.0 $^a$ |
| B20reapN2 | 18.99 ± 0.3 $^a$ | 1.34 ± 0.03 $^a$ | 1.41 ± 0.03 $^a$ | 15.44 ± 0.9 $^a$ |

Notes: Soil T–soil temperature; BD–soil bulk density; SWC–soil water content.

During the growing season of 2019, the average soil water content observed under different treatments was similar and did not vary significantly (Table 2). SWC was very low and ranged from 2 to 17% after biochar application and reapplication, with or without N-fertilizer, over the study period; however, based on these results, it cannot be concluded that biochar has no effect on soil moisture or on the crop growth during the year, as biochar is a porous material and contains micropores [70]. Biochar has the ability to retain water and improve soil structure [17,22], which can also be reflected through its improved water retention [71]; however, biochar has the ability to increase the SWC and to provide water to plants when necessary.

The soil bulk density (BD) values did not change significantly, due to either the initial application or the reapplication of biochar at any rate (Table 4), which is surprising since biochar has much lower specific density than the soil and it is expected to reduce the BD. Many studies observed decreases in BD and increases in porosity as a result of biochar application [72,73]. Roughly, 2% (by weight) of added biochar in soil is enough to show a significant decrease in BD in amended soils [72]. The rate of biochar application, its properties, as well as the density and porosity of the original soil, are critical in predicting the effects of biochar addition to any soil [73]. The differences in the BD between spring and autumn samples are natural, and are related to the changes in the volume and redistribution of pores of different sizes, the dynamics of aggregation, and the stability of the soil structure during the year [17]. Fertilization with mineral N-fertilizers can intensify the soil organic matter mineralization, which may result in an increase of soil BD. If N fertilization is combined with the application of organic amendments such as biochar, the soil structure

can be improved, and at the same time, BD [74] can be reduced, which was not observed in our experiment during the growing season of 2019.

### 3.3. Nitrous Oxide Emissions

The effect of initial application and reapplication of biochar, in both rates, on the dynamics of nitrous oxide ($N_2O$) emissions, at three different rates of N-fertilizer (N0, N1, N2), is shown in Figure 2. $N_2O$ emissions had temporal variations that were similar across all the treatments over the course of the experiment. There were increases in $N_2O$ emissions after precipitation events, especially during the spring period, and soon after the fertilizer application, whereas the impact of biochar treatments on the reduction of $N_2O$ emissions was evident. The maximum fluxes of $N_2O$, especially in N2 treatments, were measured during the period from the beginning of May to the beginning of July, with the soil temperature being between 9.3 °C and 27.2 °C, and soil moisture content between 5.0% and 25.5% of mass. Such conditions are favorable for nitrification [75,76]. The period from 5 to 24 May 2019 had 12 rainfall events which resulted in increased soil water content (22.7–25.5%), as measured on 24 May. In comparison, the SWC measured on the 26 April, before the rainfall events, ranged from 5.0% to 10.1% for all the studied treatments. The sharp increase of daily $N_2O$ emissions measured on 24 May was most likely due to high values of SWC caused by precipitation. When soils are saturated with water, the soil pores (also formed by biochar), as well as the pores of the biochar itself, are filled with water and an anaerobic environment is created. Such conditions typically lead to an increase in $N_2O$ emissions, mainly through the denitrification process [77].

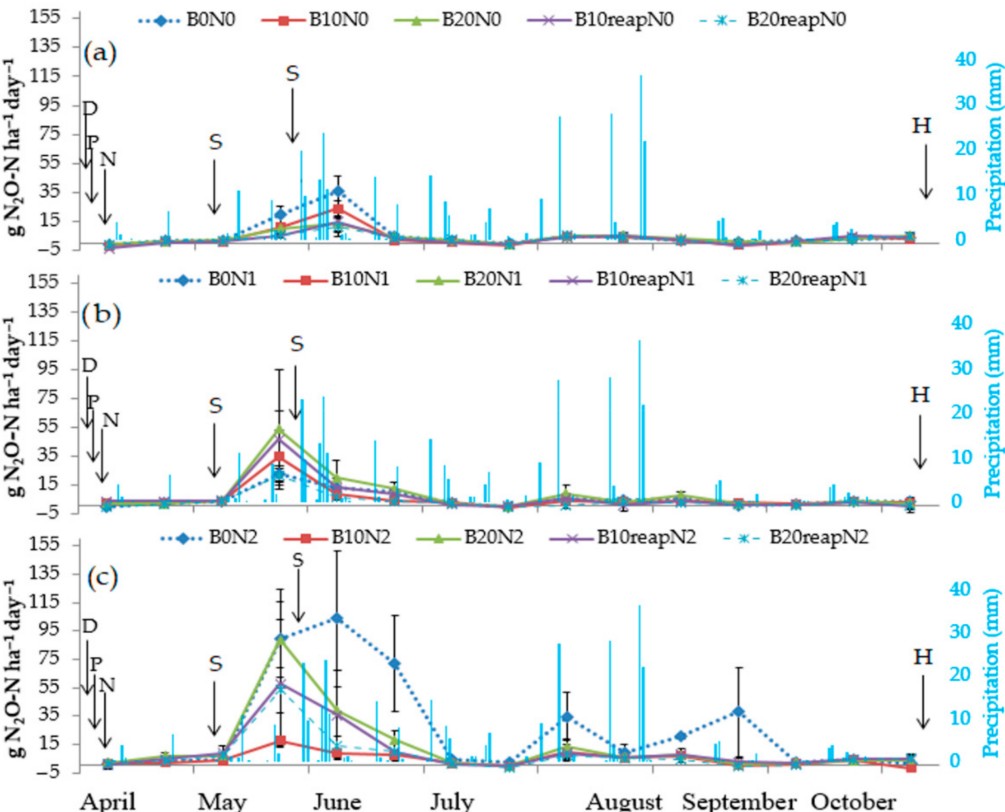

**Figure 2.** Fluxes of $N_2O$ from unfertilized soil N0 (**a**), fertilized soil N1 (**b**), and fertilized soil N2 (**c**), with or without the biochar, including precipitation. Error bars denote the standard error of the mean (*n* = 3). D—disking; P—planting of maize; N—nitrogen fertilizer application; S—spray herbicide application; H—harvesting maize.

All three biochar treatments showed different maximum $N_2O$ peaks in May and June. The significant differences were observed at fertilization level N2, where the temporal

$N_2O$ pattern was different, and fluxes were significantly higher in comparison with the fertilization levels N0 and N1. Higher $N_2O$ emissions at fertilization level N2 were also observed in the second part of July, August, and September (Figure 2c). These emission peaks were measured during the rainfall period from 27 July to 8 September with increased SWC. Generally, daily $N_2O$ fluxes from the treatments with biochar, combined with or without N-fertilizer (N2 and N0 level, respectively), were lower as compared with the control treatments (B0N2 and B0N0, respectively; Figure 2a,c) over the growing season of 2019. This was also the case when average daily $N_2O$ emissions over the whole period of the study were compared: biochar treatments combined with or without N–fertilizer (N2 and N0 level, respectively) showed lower fluxes (Figure 3a,c). Significantly ($p < 0.05$) lower average daily $N_2O$ emissions were measured in the biochar treatment without N-fertilizer (B10reapN0) and in all the biochar treatments combined with the highest fertilizer rate (N2), as compared with their control treatments (B0N0 and B0N2, respectively). This was not the case in the treatments when biochar was combined with fertilizer at the N1 level (Figure 3b). All average daily $N_2O$ emissions for all the biochar treatments at this fertilization level were not statistically different from the control B0N1 treatment (Figure 3b).

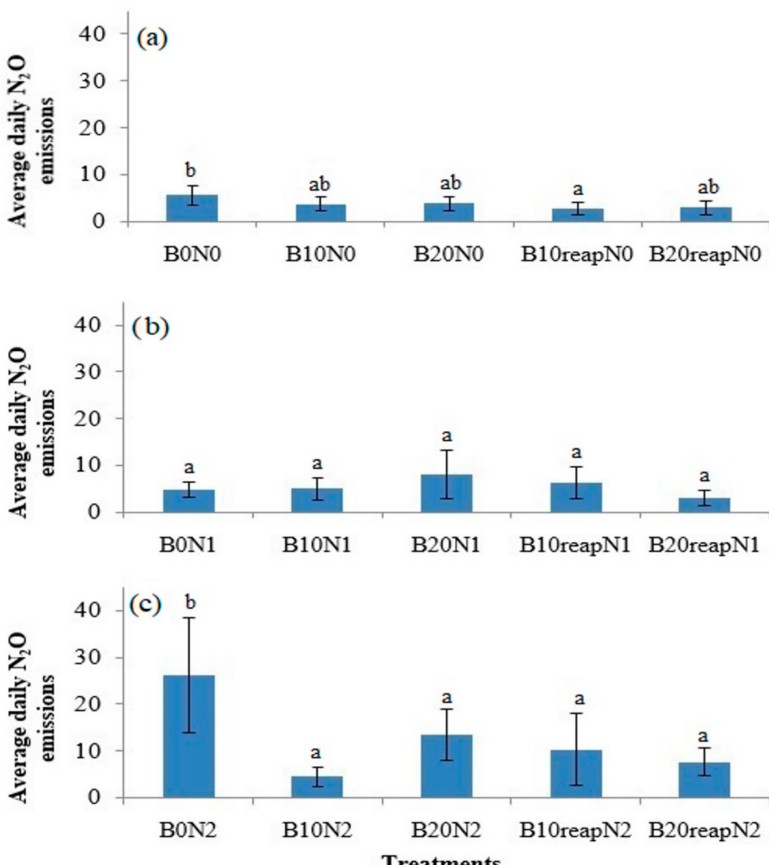

**Figure 3.** Daily average $N_2O$ emissions from unfertilized N0 (**a**), fertilized soil N1 (**b**), and fertilized soil N2 (**c**), with and without biochar. Error bars denote the standard errors of the mean ($n = 3$). Different letters in the graph indicate significant differences between different treatments, according to the Tukey procedure.

Based on the obtained results, it can be concluded that application of biochar and its combination with the lower level of N fertilizer (N1) had no effect on the elimination of $N_2O$ emissions. On the other hand, at the N2 fertilization level, a statistically significant decrease in $N_2O$ emissions was observed in all the biochar treatments, which means that biochar at this rate can eliminate $N_2O$ fluxes during more intensive nitrogen fertilization. even a long time after initial biochar application. Castaldi et al. [35] reported that higher rates of biochar have a significant impact on reduction of soil $N_2O$ emissions (by 79%)

than lower rates (by 26%). The results of the meta-analysis of 85 studies published by Shakoor et al. [78] also showed that the application of biochar amendment significantly mitigated $N_2O$ emissions by 19.7%, whereas the application rate of biochar 30 t ha$^{-1}$ had a significant effect (significantly mitigated $N_2O$ emissions by 22.5%). It is also possible that biochar has contributed to the better management of surplus nitrogen intake by maize in N2 treatments, which resulted in higher maize yields (subchapter 3.4) and reduced $N_2O$ emissions (Figure 3) during the 2019 growing season in this study. As reported by Lopez-Capel et al. [79], if biochar is reapplied to the soil, its initial effect may be reduced as result of its weak reactivity.

Multiple linear regression analysis was performed for the variables that influence $N_2O$ emissions (i.e., N-fertilization, Biochar, SWC, Soil temperature, Day (Table 5 and Figure 4) (Equation (1)):

$$\log_{10}(N_2O + const) = a_0 + a_1 \cdot N + a_2 \cdot (B + reap) + a_3 \cdot SWC + a_4 \cdot Temp + a_5 \cdot Day \quad (1)$$

**Table 5.** Multivariate linear regression coefficients and *p*–values. The dependent variable was $\log_{10}(N_2O + const)$.

| Variable | Coefficient $a_i$ | Coefficient Value | *p*-Value |
|---|---|---|---|
| Regression constant | $a_0$ | 1.0294 | <0.001 |
| N-fertilization (kg ha$^{-1}$) | $a_1$ | 0.0005 | <0.001 |
| Biochar (t ha$^{-1}$) + reapplied | $a_2$ | −0.0016 | <0.001 |
| SWC (%) | $a_3$ | 0.0088 | <0.001 |
| Soil temperature (°C) | $a_4$ | 0.0115 | <0.001 |
| Day (counted from the first measurement) | $a_5$ | −0.0003 | <0.001 |

The coefficient of determination for regression was $R^2 = 0.22$. Multiple linear regression indicates that all variables used in the analysis were statistically significant. Most of the regression coefficients were positive (i.e., emissions increase with increased nitrogen dose, soil temperature and soil moisture). The emissions decrease with time and the rate of the biochar, which confirmed our hypothesis.

In Figure 4, the surfaces show the log (N + const) regression functions versus SWC and temperature for different combinations of N and biochar. This is due to the fact that in Equation (1), log (N + const) depends significantly on the nitrogen dose (*p* < 0.001) and the use of biochar (*p* < 0.001). Regression parameters vary with the number of variables used, even for the same dataset. The highest values of the regression coefficients for soil temperature (0.0385) and soil moisture (0.0233) were calculated for the treatment B0N2. The results show that the biochar amendment has the potential to reduce the negative effect of mineral N-fertilizer on $N_2O$ emission from the soil. This is an important finding for farmers in the context of regulating nitrogen conversion processes in the soil. Although a reduction of fertilizer application is the most effective method to reduce soil $N_2O$ emissions, it comes with the cost of lower crop yields [24]. Traditional fertilization could be combined with biochar without a decrease in crop yields, according to results observed in the same field experiment [80].

Figure 5 shows that $N_2O$ emissions are strongly positively correlated with soil $NO_3^-$ and $NH_4^+$ content, and weakly positively correlated with soil temperature and SWC. $N_2O$ emission is weakly negatively correlated with soil pH. The PCA data was confirmed by the Pearson correlation analysis for the observed soil properties (Table 6).

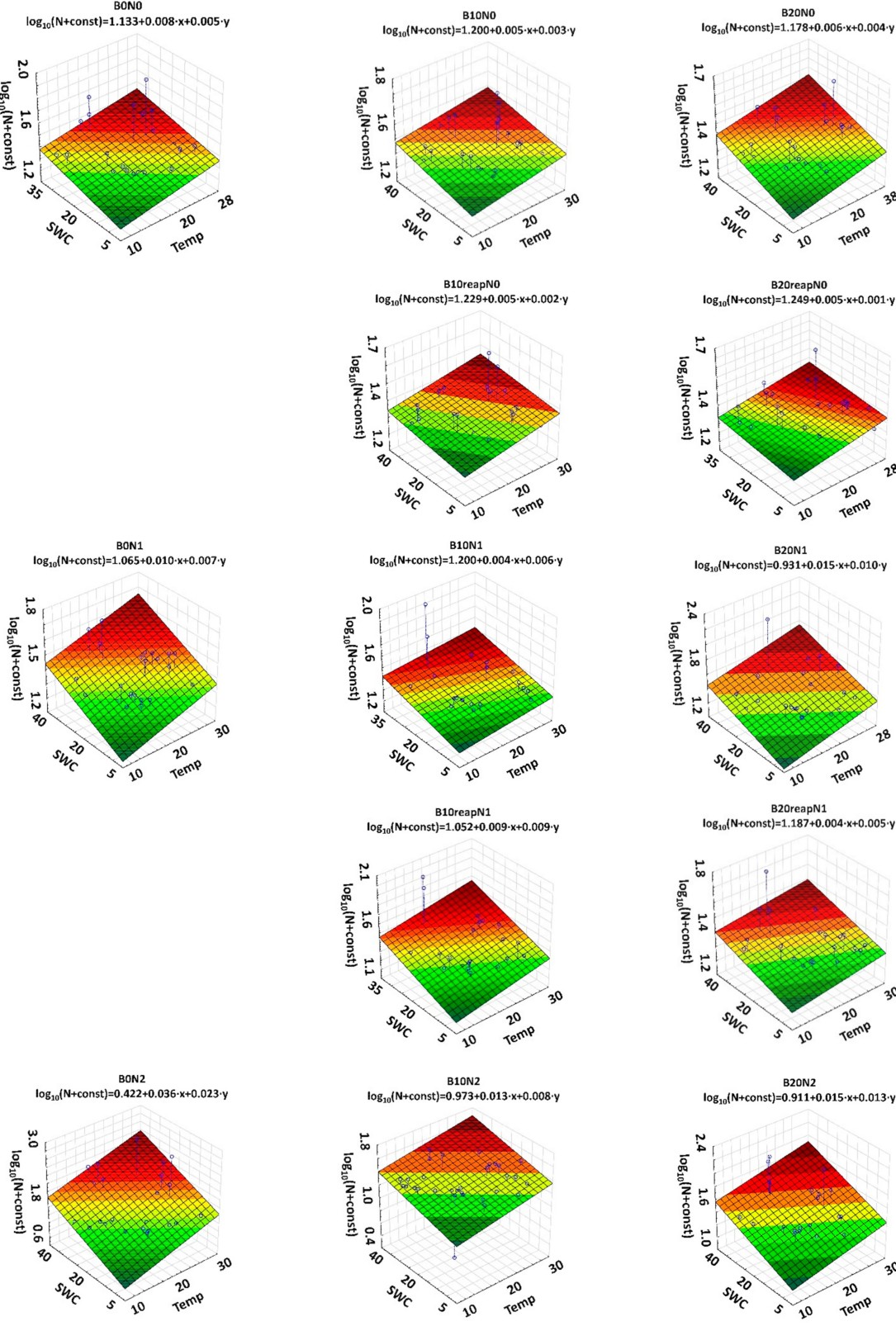

**Figure 4.** *Cont.*

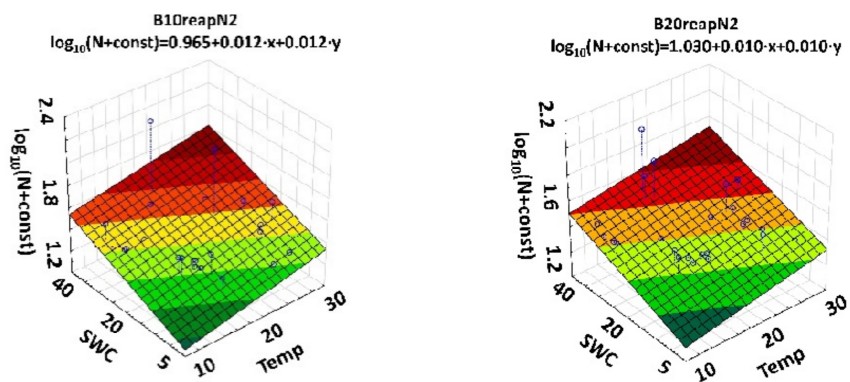

**Figure 4.** Three-dimensional plots with regression equations for the $N_2O$ emission versus soil temperature (X axis) and soil moisture (Y axis) content for all the combinations of N-fertilizer and biochar. For better visualization of the log (N + const) regression function values, they are color coded from green (lowest value) to red (highest value).

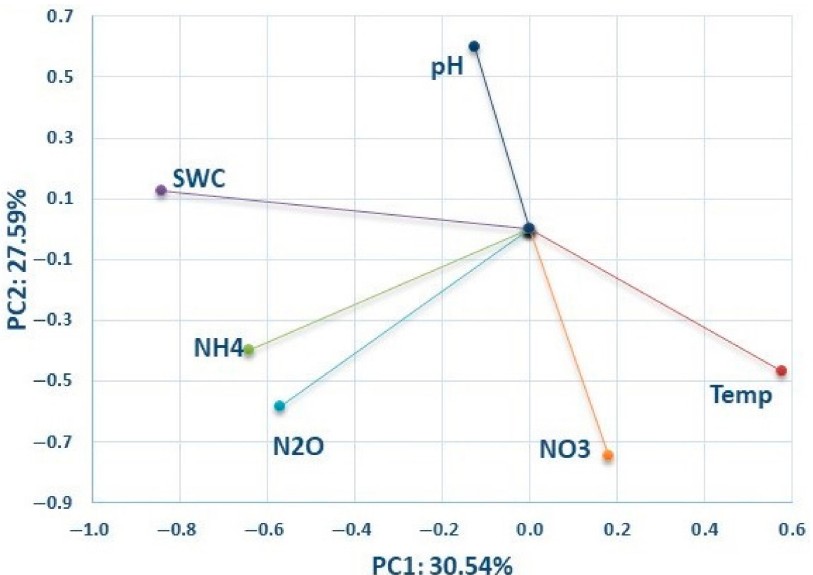

**Figure 5.** A biplot of the first two components of a PCA model of the biochar data for all dates, presenting the relationship between variables: SWC—soil water content (%), soil temperature (°C), $NO_3^-$ (mg kg$^{-1}$), $NH_4^+$ (mg kg$^{-1}$), soil pH and $N_2O$ (g ha$^{-1}$).

**Table 6.** Pearson correlation coefficients for all soil properties used in the PCA analysis.

| | N$_2$O | pH | NO$_3$ | NH$_4$ | SWC | Temp |
|---|---|---|---|---|---|---|
| N$_2$O (g ha$^{-1}$) | 1.00 | −0.25 | 0.13 | 0.39 | 0.37 | 0.06 |
| Soil pH | −0.25 | 1.00 | −0.27 | 0.05 | 0.12 | −0.12 |
| NO$_3^-$ (mg kg$^{-1}$) | 0.13 | −0.27 | 1.00 | 0.23 | −0.21 | 0.28 |
| NH$_4^+$ (mg kg$^{-1}$) | 0.39 | 0.05 | 0.23 | 1.00 | 0.31 | −0.15 |
| SWC (%) | 0.37 | 0.12 | −0.21 | 0.31 | 1.00 | −0.41 |
| Temp (°C) | 0.06 | −0.12 | 0.28 | −0.15 | −0.41 | 1.00 |

Notes: SWC–soil water content; Temp–soil temperature. The statistically significant correlations are shown in bold font. Color gradient stands for the strength of the observed correlations and it ranges from dark red (strong negative correlation) to dark blue (strong positive correlation).

*3.4. Maize Yield*

The maize yield at the experimental site ranged from 6.4 t ha$^{-1}$ to 12.8 t ha$^{-1}$ (Figure 6). When comparing the observed values with the national average maize yield (7.33 t ha$^{-1}$ in 2019) according to public database of Statistical Institute of Slovakia [81], the

obtained yields were above the national average (except for treatment B20reapN0). This is an interesting observation considering that the climatic conditions at the experimental site were not favorable, especially at the beginning of the vegetation period when, after the dry April when the crop was sown, the maize was growing in the conditions of extremely wet and extremely cold May.

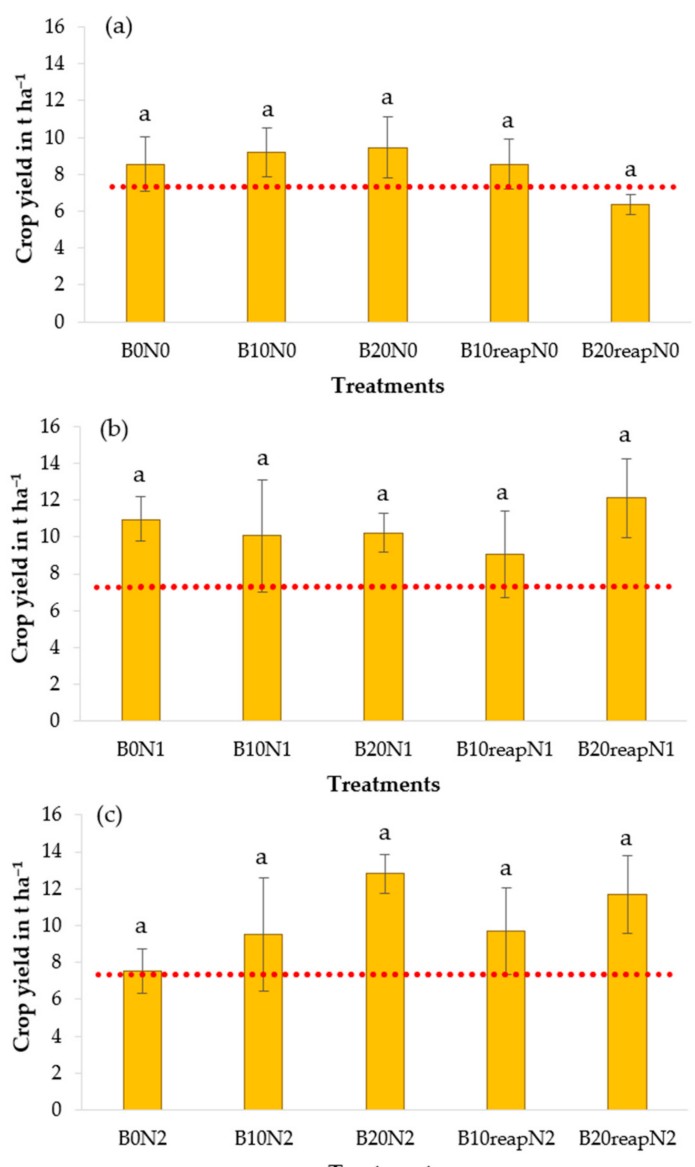

**Figure 6.** Maize grain yield in the treatments with and without biochar at unfertilized level N0 (**a**), and fertilized levels N1 (**b**) and N2 (**c**). Error bars denote the standard errors of the mean (*n* = 3). Different letters in the graph indicate significant differences between different treatments according to the Tukey procedure (*p* < 0.05). Dotted line delineates the average maize yield in Slovakia in 2019.

However, according to our results, application or reapplication of biochar did not have a significant effect on maize yield at the studied treatments in 2019. The observed trends differed at the N fertilization levels; however, due to a rather high variability of data within treatments, the general description of the trends in the yield is just an assumption. At N0, the maize yield only slightly increased (by 7% and 11% for B10N0 and B20N0) after a single biochar application. Reapplication of biochar did not result in an expected increase of yield compared with a single biochar application. Similarly, when biochar was combined with fertilizer at N1, no effect of biochar was observed. In treatment B20reapN1, the maize yield

was higher by 11%; on the other hand, in treatment B10reapN1, the observed yield was lower by 21% compared with the control treatment B0N1. At the fertilization level with the higher fertilizer input (N2), all treatments with biochar resulted in an increase of crop yield ranging from 27% to 71% ($p > 0.05$) when compared with the control treatment B0N2. At the same time, the increasing trend was more visible in treatments with biochar application or reapplication at a dose of 20 t ha$^{-1}$.

## 4. Conclusions

The initial application and reapplication of biochar, as well as biochar combination with the higher level of N-fertilizer, had positive effects on improving the soil pH, in the range of 7–13%, thus increasing soil organic carbon by 2–212%, reducing ammonium in the range of 41–69%, and reducing N$_2$O emissions from the soil in the range of 33–83%. Conversely, a lower level of N-fertilizer combined with the biochar had no effect on the changes in the soil properties and on the reduction of N$_2$O emissions from the soil to the atmosphere. N$_2$O emissions were always increasing after precipitation events, especially in the spring and summer. In general, the increasing nitrogen rate, temperature, and soil moisture resulted in increased N$_2$O emissions, whereas, on the other hand, emissions decreased over time because of biochar application—the effect was more pronounced at a higher application rate. Application or reapplication of biochar did not have effect on maize yield in 2019. The results of this study suggest that in order to improve soil properties and maintain lower N$_2$O emissions, it is extremely important to apply N-fertilizer in combination with biochar, particularly at higher fertilizer rates. Attention and extra study are required to investigate why biochar did not have such an effect at a lower N-fertilizer dose. The change in soil properties and relationships between them are a result of the initial application and reapplication of biochar as well as biochar application in combination with N-fertilizer, which was reflected on the soil N$_2$O emissions. The results indicated that the main reasons for the reduction of N$_2$O emissions in biochar treatments, and treatments where biochar was combined with N-fertilizer, were the reduction of mineral nitrogen, the increase of soil pH, and the increase of the soil organic carbon content through the application of the biochar to the soil.

In the future, it is extremely important to pay attention to the pairing of these results with the yields of crops, preferably for the entire duration of the experiment, as it is only on the basis of multi-year results that it is possible to make more relevant proposals/recommendations for agronomy and ecology in the context of biochar and the soil and climatic conditions of Central Europe.

**Author Contributions:** Conceptualization, T.K., J.H. and V.Š.; methodology, J.H. and T.K.; formal analysis, T.K., J.H., K.D., E.W.-G. and V.Š.; investigation, T.K., J.H., K.D., E.A. and V.Š.; resources, T.K., J.H., K.D. and M.I.; data curation, T.K., K.D., E.W.-G., E.B., N.B., M.I., E.A. and J.H.; writing—original draft preparation, T.K., J.H. and V.Š.; writing—review and editing, T.K., J.H., E.B., N.B., V.Š. and E.A.; visualization, T.K. and E.W.-G. and E.A.; supervision, J.H. and V.Š.; project administration, J.H.; funding acquisition, J.H. All authors have read and agreed to the published version of the manuscript.

**Funding:** This study was supported by the Slovak Grant Agency (VEGA) project no. 1/0116/21 and 1/0021/22, the Cultural and Educational Grant Agency, KEGA (No. 019SPU–4/2020), and the Operational Program Integrated Infrastructure within the project "Sustainable smart farming systems, taking into account the future challenges 313011W112", co-financed by the European Regional Development Fund. Drs. E.V. Balashov and N.P. Buchkina were partly working according to the scientific topic of the Agrophysical Research Institute.

**Institutional Review Board Statement:** Not applicable.

**Informed Consent Statement:** Not applicable.

**Data Availability Statement:** The datasets generated during and/or analysed during the current study are available from the corresponding author on request.

**Conflicts of Interest:** The authors declare no conflict of interest. The funders had no role in the design of the study; in the collection, analyses, or the interpretation of the data; in the writing of the manuscript, or in the decision to publish the results.

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
