# Peer review of "Combination of Biochar with N–Fertilizer Affects Properties of Soil and N2O emissions in Maize Crop"

_agronomy, doi:10.3390/agronomy12061314_

Round 1

Reviewer 1 Report

General comment

This is a nice study on the effect of different rates of biochar amendments combined with several N fertilizer levels on soil properties and N2O emissions. The experiment was performed during a cycle of maize crop cultivation under field conditions, which give more insight of the potential benefits of biochar application in real crop conditions. However, I believe that yield crop data or, at least, crop N uptake should have been included.

Another concern is that the authors already published a similar manuscript from this experimental site with data from the previous season crop cycle. To my opinion, the authors should support the relevance of this research by highlighting what is novel in this new publication. I believe that comparisons between these two studies could be of high interest. Maybe some observed effects are supported or in contrast in this new season, as similar parameters were measured a year later in interaction with a distinct crop.

I miss some discussion about the effects of biochar dosages. From the information depicted in tables and figures it seems that there were not differences and a unique application of the lower rate of biochar could be advisable.

Specific comments

Lines 141-157: The authors could have calculated the soil WFPS, which is more common in studies on soil N processes and related N2O emissions.

Lines 203-235: Discussion about mineral N retention by biochar surface rely on previous studies but I am not sure whether the results of this experiment are consistent with these hypotheses. In my opinion, with this experimental design, it is not possible to assess if biochar had some effect in retaining NH4+ or NO3- (via absorption or adsorption), or favoured microbial activity or nutrient availability for plant uptake.

Lines 236-242: I am not sure about the message the authors want to give. Is it necessary to combine N fertilisation with biochar application for C sequestration? Or is it that there were not remarkable results due to field heterogeneity?

Lines 260-263: Maybe it is not necessary to include this information within the text. There were not differences between treatments and the numbers are already in table 4.

Line 273: Table 4. I believe this paragraph repeats information from the previous one.

Lines 280-285: This also affect N uptake. I believe that yield information should be included in this manuscript.

Lines 287-289: Biochar application rate was similar to the rates tested in this experiment? What about differences or similarities with biochar and soil properties?

Line 314: The link to this reference is broken and the manuscript is written in Chinese. Maybe this sentence could be deleted, as the next one sounds repetitive and includes other references about the same mechanism.

Line 352: Is this a result from your study or from the referred one? Results from N1 treatment differed from N2 level. The authors argument at the end of this paragraph that the presence of biochar in the presence of N surplus could have mitigated N2O emissions… Please, clarify

Line 354: The referred study is about a compost matrix, which is quite different to the conditions of this experiment.

Lines 364-380: Were the observed differences significant? I believe statistics from cumulative N2O emission would be very similar to those of daily average emissions shown in figure 3. According to the results depicted in the figure, there were not significant differences between treatments N1 (which is already in contrast with N0 and N2 results). As this data is not shown and repeats part of the given information in the previous paragraph, I think this paragraph could be deleted.

Line 386: Clarify units for equation 1

Lines 394-395: Was it also confirmed that temperature and water content were the most significant variables affecting N2O emissions? It is possible that N dosage or the interaction between N and biochar rate were even more relevant.

Lines 398-399: Explain the meaning of the colour gradient in figure 4.

Line 401: The equation within figure 4 shows other number.

Lines 405-408: This study does not cover the impacts on crop yields.

Line 419: Revise this. In other parts of the manuscript N2O emissions are reported positively (not negatively) correlated with temperature and water content (emissions increase when temperatures and SWC are high in figure 2)

Line 445-447: I am not sure that this is supported by the results from this study. High pH favours the last step of denitrification, which mitigates N2O emissions, but there could be other variables with a higher impact. Moreover, according to table 6, ammonium soil concentration was the most significant impact on N2O emissions during this cropping period.

Lines: 471-475: Why the authors did not relate this study with their previous results from this experiment? I think that a comparison, including crop yields and/or crop N uptake, as well as the evolution of the studied parameters would have raised the novelty and interest of this manuscript.

Revise the format of the references. Some links are broken.

Author Response

Assoc. prof. Dr. Ján Horák

SAU Nitra, 4.5.2022

Dear Editorial Board,

Please find enclosed revised manuscript after the second review round entitled: „ Combination of Biochar with N–fertilizer Affects Properties of Soil and N2O emissions" which I am submitting for consideration of publication as a “original research in the Special Issue "Black carbon and its effects on crop productivity and nutrient cycling” of Agronomy

We have significantly and deeply revised the manuscript in a second round, and we think that it can be considered for publication in Agronomy journal, even we were not able to fulfil all requirements of the one reviewer. However, we think that we fulfilled all requirements of other two reviewers.

Thank you for your consideration of our study. Please address all correspondence concerning this manuscript to me at Slovak University of Agriculture in Nitra and feel free to correspond with me by e-mail (jan.horak@uniag.sk).

Yours sincerely

Ján Horák

Reviewer 2 Report

Dear Authors

The current study entitled “Combination of Biochar with N–fertilizer Affects Properties of Soil and N2O emissions” is good. For a better understanding in-depth, it is a need for time to work on this topic. Furthermore, the achievement of potential benefits by using current technology is also dependent on the extensive research work for more exploration. Although the experiment is well organized, I suggest a minor revision due to the following deficiencies.

Abstract

  1. Make the title a simple statement.
  2. Quantitative data is also important to support your conclusion. Would you please provide some quantitative data in terms of percentage significant increase or decrease in the abstract?
  3. Please provide a conclusive conclusion with is withdrawn through research in a single line.
  4. Give future prospective in a single line.
  5. As per standard suggestions, please avoid using title words as keywords.

Introduction

  1. Also, provide a novelty statement at the end. What new things authors have done or correlated in this research compared to old ones?
  2. Would you please give a single line about the knowledge gap which your research has covered along with the hypothesis statement?

Material and methods:

  1. Please provide a reference for statistical analysis.

Results and Discussion.

  1. These sections are fine.

Conclusion

  1. Add the targeted beneficiary audience who will get benefits from this research.
  2. Also, give clear-cut recommendations while describing the best treatment.

Author Response

(The authors gave the same response as above.)

Reviewer 3 Report

The manuscript entailed- “Combination of Biochar with N–fertilizer Affects Properties of Soil and N2O emissions “is critically reviewed again and found that the authors have improved the manuscript significantly as over previous submitted one and therefore I strongly recommended this study for acceptation and future consideration in Journal agronomy. 

Author Response

(The authors gave the same response as above.)

Round 2

Reviewer 1 Report

I find little discussion comparing the single dose of biochar with its reapplication. Many of the depicted results show no significant differences between these two strategies. The authors conclude that both strategies are valid (also with no differences)

I believe the manuscript is incomplete without the information about crop N uptake and yield. If the authors have this information, it is not justified not to include it just because they have to perform the statistics

Author Response

Response to Reviewer 1 Comments

Thank you very much for your involvement in the revision process and the time that you dedicated to assessing our manuscript and for the great job done. Hopefully we have met all your expectations which we were able to do. Thank you very much for improving our manuscript.

 Comments and Suggestions for Authors

I find little discussion comparing the single dose of biochar with its reapplication. Many of the depicted results show no significant differences between these two strategies. The authors conclude that both strategies are valid (also with no differences)

Response: Thank you very much for your comments. We have revised and added discussion comparing the single dose of biochar with its reapplication.

I believe the manuscript is incomplete without the information about crop N uptake and yield. If the authors have this information, it is not justified not to include it just because they have to perform the statistics

Response: Thanks for your comment. We have only complete data about the mineral N (NH4+, NO3-), which are detailed described in the  manuscript. Unfortunately, we do not have data about crop N uptake, so we are not able to process it. We didn’t do the N-labeling. Thank you very much for understanding. We have processed data on crop yields and we added these results of crop yields in our manuscript. We also added another co-author into the manuscript who is in charge of the yields in our experiment and who has done the evaluation and writing in this manuscript. 

This manuscript is a resubmission of an earlier submission. The following is a list of the peer review reports and author responses from that submission.

Round 1

Reviewer 1 Report

General comment

This work studies biochar impact on soil properties and N2O emissions in a field experiment during a cycle of maize crop cultivation. The paper is well written and provides additional evidence of the benefits of applying biochar to acid agricultural soils in warm temperate climates.

I find some results contrasting and I am not sure of the suitability of the statistical analyses (this is more detailed in the specific comments)

Furthermore, some parts of the discussion seem to me rather speculative. Arguments are often based on analyses from other studies that were not performed in this experiment.

Specific comments

Title: I am not sure about the message that the authors want the readers to understand, in special with “different ranges on N2O emissions”

Introduction

Lines 39-42: I do not see the point of including here such a wide spectrum of agricultural practices (i.e. livestock farming). I also believe that other reference would be better fitted here (the authors could use others already included in the reference list)

Lines 44, 45: I believe some reference supporting this information should be included here.

Lines 54-57: I suggest the authors to focus on practices related with this study.

Lines 63-66: I believe it is not necessary to include this high number of references here. Additionally, many of them belongs to the same authors...

Lines 67-85: I find this paragraph about N2O emissions from agricultural soils and the observed effects of biochar amendments in previous studies ok although rather incomplete… Most studies report reductions of N2O emissions after biochar amendments, but results sometimes are contrasting depending on the type of experiment (laboratory soil incubations, pot experiments or field experiments), the duration (short, mid or long-term experiments) the type of soil and biochar, or the driving soil mechanism leading to N2O formation and release. There are metanalyses dealing and summing up previous evidence of these diverse observed effects.

Materials and methods

Line 102: …to a temperate…

Line 106: Delete the minus sign (-) before 10.9

Line 108: Average of?

Line 109: Delete “-“ and include wether pH was measured in KCl and the extract ratio.

Table 2: Include KCl extract for pH determination and the technique used to determine specific surface area. I would also include molar H/C and O/C ratios of the biochar.

Lines 127-140: I suggest the authors including the crop history of the experimental plot. In line 138: “Figure 1 shows…

Lines 146-159: It would be more useful including the Water Filled Pore Space (WFPS, rather than the gravimetric water content). It is very easy to calculate as the authors have also determined the BD of the soil.

Line 163: Sampling every two weeks may be ok when emissions are low and almost constant. However, it would had been advisable measuring N2O emissions more often after precipitation events, at least when they took place in combination with high temperatures, to properly register the evolution of the flux peaks.

Line 164: What metal? May it interfere with redox reactions involved in the N2O formation and release?

Line 169: Water sealing?

Lines 178-185: To my view, a two-way ANOVA with N and biochar rates as factors would be better suited.

Results and Discussion

Lines 189-204: This paragraph about soil pH is quite large but its connection with N2O emissions is missing.

Line 205: After each N-fertilization? In table 3 there is only one point value.

Line 215: Although the experiment was stablished in a maize crop field, interactions with plants were not measured.

Line 218: Neither CEC was measured

Lines 222-223: Similar comment as pervious. Several mechanisms are simultaneously happening in the soil matrix and with this experimental design it is not possible to unravel the biochar impact on specific mechanisms such as the nitrification process or N plant uptake.

Lines 228-239: I would revise this paragraph as some of the arguments seem speculative (i.e., biochar elimination of nitrification elimination, impact of its physical properties, NO3- adsorption or NH4+ retention). What is the evidence provide by this study that supports this part of the discussion?

Lines 246-256: Is it possible that the observed differences between Corg of spring and autumn samples were a consequence of field heterogeneity? They do not seem significant.

Table 2: Indicate pH dilution. I believe a two-way anova would give more information about biochar and N-fertilization impact on soil properties.

Line 264: Delete “-“

Line 267: I suggest including the WFPS of the soil. It is more intuitive to identify the possible predominant mechanisms operating in the soil.

Line 276: Reference #62 is not about biochar

Line 279: In reference #63, Jin et al. study the effect of pyrolysis temperature on sewage sludge derived biochar, including heavy metals availability. However, the referenced study does not provide information about the effect of soil moisture or the presence of vegetation.

Lines 281-282: This was said in lines 266-267

Lines 284-290: These are observations from other studies but not from this study.

Line 293: Why the authors do not include in this manuscript information about the maize crop performance during this study?

Lines 304-305: I do not fully understand why soil structure is improved while its BD is reduced (soil compaction) at the same time. Moreover, I do not see the link with your results. It is difficult to see significant impacts on soil structure when there is a high field variability.

Table 4: See previous comments about statistical analyzes and WFPS

Lines 336-341: It is possible that flux patterns differed between treatments, but it would had been necessary measuring gases more frequently to register this possible effect.

Line 337: “nitrous oxide”

Line 344: Delete “-“. I suggest again using WFPS.

Lines 344-353: Several mechanisms, mainly nitrification, nitrifier denitrification and denitrification, use to operate at the same time within the soil matrix and the N2O emission increase with soil moisture and temperature is consistent with previous studies. Maybe the authors could compare their N2O emissions with similar studies using biochar amendments.

Figure 2: I would include soil temperature and WFPS instead of precipitation. I suggest increasing font sizes of all the figures of the manuscript

Lines 361-363: This sounds repetitive, already said in the previous paragraph. If the authors included results from crop performance (such as yield or N uptake), it could be possible to relate the highest N2O emissions registered in N2 treatment with a possible N surplus.

Lines 375-378: This is hard to follow… Additionally, these differences were not significant

Lines 381 and 386-388: These sentences are contradictory. Moreover, I do not think that the reported effect from lines 382-384 from other studies explains the observed patterns in this study. Why did biochar pores clog at N1 fertilization level but not at N2?

Lines 394-407: Where is this information depicted? Were these increases/reductions of cumulative N2O emissions significant? According to the information shown in figure 3 (average daily emissions), most of these differences were not significant.

Lines 411-419: In the equation, indicate the meaning of the abbreviations and identify each ai coefficient. It would help readers if they appeared in the equation and in the table in the same order.

Line 421: The value of R2 is quite low…

Lines 424-425: But your previous results do not depict a significant impact of biochar rate.

Line 435: Was this also observed in your study?

What new information provide all these correlations? The positive correlation between temperature, moisture and N2O emissions was already said

Line 448: Delete “-“. This is in contrast with previous information contained in this manuscript as well as with previous studies. Why do the authors believe this is due to? What are the causes of the different trends observed in the two studied dates?

Line 459: Delete “-“

Lines 470-486: I do not fully understand the purpose of performing two PCA analyzes (one for spring and another for autumn). The results are contrasting and not properly explained (also contrasting with correlations previously found). What value of N2O emissions was introduced for the analyses? If it was cumulative, cumulative emissions in autumn includes the spring period. Please, clarify. I believe nitrification and denitrification processes were responsible of N2O emissions during the whole experiment (not just in autumn, line 475). To my view, these results do not support the theory of biochar nutrient adsorption or availability from biochar labile mineral N content (not measured)

Conclusions

Line 497: Delete “-“

I believe stronger conclusions could be made if the manuscript included crop performance

Reviewer 2 Report

The manuscript entailed- “Combination of Biochar with N–fertilizer Affects Properties of Soil with Different ranges on N2O emissions “is critically reviewed and found to be a very interesting study. This manuscript covers the important nitrous oxide aspects in the agricultural system. This study has global significance in the present climate change and global warming era and therefore I strongly recommended this study for future consideration in Journal agronomy with following suggestions given below:

Abstract: The abstract of a good manuscript always has quantitative finding, which is completely lacking in this study. Authors are   suggested to add quantitative findings in abstract and abstract should we end up with future recommendations. Add both, quantitative finding and future recommendation, as they are important part and missing in abstract.  

Try to avoid writing long sentences (two lines or around 30 words): Long sentences reduce the readability of the reader. Split all long sentences in to two or required number of sentences 

Line no: 39-42: suggested to add https://doi.org/10.1007/s11356-021-14210-z along with the references no , 3-4.

Line no: 42-44- Support sentence with relevant reference

Line no: 59: but also a significant factor that can reduce the production of GHGs through various mechanisms ------suggested to support with https://doi.org/10.1016/j.rser.2021.111379  

Section 2.5: Suggested to write about the equation used for the calculation of N2O emission and write briefly about interpolation methodology. Kindly support it with relevant references also.

Figure no 2: Kindly improve the quality of image, if possible

Figure no 3a and 3b: Limit the y-axis range to 20 only, it will improve the visibility of the Figure.

Figure no 4: Please cross the image, are all plots in figure???

Conclusion:  Add quantitative finding and one-two line about future recommendation.

Reviewer 3 Report

The current study entitled “Combination of Biochar with N–fertilizer Affects Properties of Soil with Different ranges on N2O emissions” is good. For a better understanding in-depth, it is a need for time to work on this topic. Furthermore, the achievement of potential benefits by using current technology is also dependent on the extensive research work for more exploration. Although the experiment is well organized, yet I suggest a major revision due to the following deficiencies.

Major Concerns

  • Systematic abstract is missing. Introduce the need for study in 1-2 lines.
  • Please give a clear-cut point problem source as a problem statement that is tackled in the current study.
  • Give logical reason for the selection of current strategy i.e., Biochar with N–fertilizer.
  • Quantitative data is also important to support your conclusion. I request the authors please carefully check and rewrite the results part in the abstract. Please provide a percentage increase or decrease in the result part.
  • Please provide a conclusive conclusion that is withdrawn through research in a single line.
  • Please conclude with a statement that shows a knowledge gap covered, potential beneficiaries and specific recommendations as well.
  • Give future prospective in a single line. At least declare one best treatment i.e., biochar with N or without N. Please do not make vague statements as in science no space for that.
  • As per standard suggestions, please avoid using title words as keywords
  • Please follow the title in the introduction section, i.e., Biochar, N–fertilizer, properties of soil affected by N and biochar, knowledge gap, hypothesis and aims.
  • Also, provide a novelty statement at the end. What new things authors have done or correlated in this research compared to old ones?
  • Would you please give a single line about the knowledge gap which your research has covered along with the hypothesis statement?
  • Material and methods: Please provide references, which methods were used for the analysis of soil characteristics.
  • Results: Give quantitative data in terms of percentage increase and decrease.
  • Give details of treatment below the tables. All tables should be self-explanatory.
  • Discussion: Please provide definite mechanisms associated with the improvement of growth attributes.
  • Add the targeted beneficiary audience who will get benefits from this research.
  • Also, give clear-cut recommendations
  • Give future prospective regarding this research.